# QUIRMIA—A Phenotype-Based Algorithm for the Inference of Quinolone Resistance Mechanisms in *Escherichia coli*

**DOI:** 10.3390/antibiotics12071119

**Published:** 2023-06-28

**Authors:** Frank Imkamp, Elias Bodendoerfer, Stefano Mancini

**Affiliations:** Institute of Medical Microbiology, University of Zurich, 8006 Zurich, Switzerland

**Keywords:** quinolone resistance, *E. coli*, whole genome sequencing, diagnostic algorithm, phenotype/genotype correlation

## Abstract

Objectives: Quinolone resistance in *Escherichia coli* occurs mainly as a result of mutations in the quinolone-resistance-determining regions of *gyrA* and *parC*, which encode the drugs’ primary targets. Mutational alterations affecting drug permeability or efflux as well as plasmid-based resistance mechanisms can also contribute to resistance, albeit to a lesser extent. Simplifying and generalizing complex evolutionary trajectories, low-level resistance towards fluoroquinolones arises from a single mutation in *gyrA*, while clinical high-level resistance is associated with two mutations in *gyrA* plus one mutation in *parC*. Both low- and high-level resistance can be detected phenotypically using nalidixic acid and fluoroquinolones such as ciprofloxacin, respectively. The aim of this study was to develop a decision tree based on disc diffusion data and to define epidemiological cut-offs to infer resistance mechanisms and to predict clinical resistance in *E. coli*. This diagnostic algorithm should provide a coherent genotype/phenotype classification, which separates the wildtype from any non-wildtype and further differentiates within the non-wildtype. Methods: Phenotypic susceptibility of 553 clinical *E. coli* isolates towards nalidixic acid, ciprofloxacin, norfloxacin and levofloxacin was determined by disc diffusion, and the genomes were sequenced. Based on epidemiological cut-offs, we developed a QUInolone Resistance Mechanisms Inference Algorithm (QUIRMIA) to infer the underlying resistance mechanisms responsible for the corresponding phenotypes, resulting in the categorization as “susceptible” (wildtype), “low-level resistance” (non-wildtype) and “high-level resistance” (non-wildtype). The congruence of phenotypes and whole genome sequencing (WGS)-derived genotypes was then assigned using QUIRMIA- and EUCAST-based AST interpretation. Results: QUIRMIA-based inference of resistance mechanisms and sequencing data were highly congruent (542/553, 98%). In contrast, EUCAST-based classification with its binary classification into “susceptible” and “resistant” isolates failed to recognize and properly categorize low-level resistant isolates. Conclusions: QUIRMIA provides a coherent genotype/phenotype categorization and may be integrated in the EUCAST expert rule set, thereby enabling reliable detection of low-level resistant isolates, which may help to better predict outcome and to prevent the emergence of clinical resistance.

## 1. Introduction

Quinolone resistance in *Escherichia coli* can be attributed primarily to the stepwise accumulation of mutations in the quinolone-resistance-determining regions (QRDR) of the *gyrA* and *parC* genes coding for the target enzymes: DNA gyrase and topoisomerase IV [1]. Decreased permeability resulting from reduced production of outer membrane porins or overexpression of efflux pumps may also contribute to quinolone resistance, albeit to a lesser extent [1]. Furthermore, a number of plasmid-mediated quinolone resistance (PMQR) mechanisms have been recently reported: efflux pumps (QepA, OqxAB) [2,3], quinolone-modifying enzymes (acetyltransferase AAC(6′)-Ib-cr) [4] and protective proteins (QnrABCDS) [5,6,7].

The first step in the evolution of clinical quinolone resistance is the selection of a mutation in *gyrA* (S83L). This initial event causes low-level fluoroquinolone resistance, which can be easily detected phenotypically by disc diffusion using nalidixic acid [8]. A mutation in *parC* (S80I) and a second mutation in *gyrA* (D87N) represent major second and third steps in clinical quinolone resistance development. The simultaneous presence of two mutations in *gyrA* plus one additional mutation in *parC* has been shown to have a limited fitness cost and to produce an epistatic effect; quinolone resistance in triple mutants is significantly higher than the sum of the respective resistance levels conferred by each single mutation alone [9]. Chromosomal mutations leading to altered drug permeation and/or increased efflux, such as, e.g., in *marR* or *acrR*, mainly occur late in resistance evolution and are associated with a high fitness cost and comparably low levels of resistance [10,11,12]. In addition to an evolutionary scenario driven by chromosomal mutations, PMQR provide a favorable background for the further selection of mutations in the QRDRs [13].

In recent years, quinolone resistance in *E. coli* has steadily increased worldwide, largely due to improper use of these antibiotics in a variety of clinical settings [14]. In Switzerland, the number of fluoroquinolone-resistant invasive *E. coli* isolates has almost doubled from 10.3% to 20.3% between 2004 and 2017 [15]. In 2019, most of the northern European countries reported resistance rates for *E. coli* between 10–25%, while various southern and eastern European countries observed rates as high as 50% (http://atlas.ecdc.europa.eu/public/index.aspx, accessed on 15 May 2019). Still, fluoroquinolones remain important first and second line treatment options for a large spectrum of diverse clinical indications, such as uncomplicated urinary-tract infections [16], prostatitis [17], pyelonephritis [18], abdominal infections, such as pancreatitis, and peritonitis or community-acquired pneumonia [19]. In addition, fluoroquinolones are used for prophylaxis prior to trans rectal ultrasound (TRUS)-guided biopsy [20] or dental procedures.

Since 2010, EUCAST no longer advises the use of nalidixic acid to screen for low-level quinolone resistance in *Enterobacterales*. In most diagnostic laboratories, susceptibility towards fluoroquinolones is deduced from ciprofloxacin. Based on the inhibition zone diameter values for ciprofloxacin and clinical breakpoints of 22 mm and 25 mm, isolates are categorized as “resistant” or “susceptible” to all fluoroquinolones [21]. One consequence of discontinuing the screening with nalidixic acid is that clinical isolates with low-level quinolone resistance (mostly resulting from one point mutation in the QRDR) may be either classified as susceptible or resistant.

Due to significantly reduced costs and shorter turnaround times, whole genome sequencing (WGS) has become a valuable technology in recent years, allowing large-scale analyses of bacterial isolates, including molecular typing and identification of genes and/or mutations responsible for antibiotic resistance. Yet, implementation of WGS in routine diagnostics as a means of genetic antimicrobial susceptibility testing is still hampered by various limitations, above all by the extensive lack of experimental evidence linking genotypic and phenotypic data. In the study presented here, we investigated the correlation between disc diffusion and genotypic data and explored the potentials and limitations of phenotype- and genotype-based inference of quinolone resistance in *E. coli* by deploying different algorithms. In general, such an algorithm should provide coherent genotype/phenotype categorization. Genetically, the wildtype is defined as absence of any acquired resistance mechanism, in contrast to the non-wildtype, which is characterized by the presence of different resistance determinants. Thus, in the interest of a coherent genotype/phenotype categorization, which has become more important since the advent of whole genome sequencing, a diagnostic algorithm requires two steps: first, separation of wildtype from the non-wildtype; second, a further differentiation of the latter to detect clinical resistance [22].

Based on the analysis of a collection of 553 clinical *E. coli* isolates, we developed QUIRMIA (Quinolone Resistance Mechanisms Inference Algorithm), a dichotomic decision tree for the inference of quinolone resistance mechanisms from inhibition zone diameters for nalidixic acid and norfloxacin.

## 2. Results

### 2.1. Design of the Quinolone Resistance Mechanisms Inference Algorithm (QUIRMIA)

We initially sequenced the genomes of 553 clinical *E. coli* isolates. Based on the detected resistance mechanisms and information derived from the literature, the isolates were classified as wildtype (WT), low-level (LLR) or high-level resistant (HLR) (see Appendix A). In 140/553 isolates, no mechanism conferring quinolone resistance was detected and these were genetically categorized as WT. Out of the 142 isolates genetically categorized as LLR, the vast majority (125/142, 88%) carried a single *gyrA* mutation, which was predominantly located in codon 83 (118 of 125 isolates). Furthermore, 24 of the isolates with a single *gyrA* mutation additionally harbored either acquired quinolone resistance mechanisms (10/24) that are usually located on plasmids (hereafter referred to as PMQR), a single mutation in the *parC* gene (9/24) or multiple mutations in the gene coding for MarR (5/24), which regulates the expression of various efflux pumps [23]. Finally, 4/142 isolates carried two *gyrA* mutations, and in 13/142 (9.1%) isolates, only PMQR were detected. In all 271 isolates categorized as HLR, two *gyrA* mutations plus at least one *parC* mutation were detected. In all but one isolate, the two *gyrA* mutations were in codon 83 and 87. Regarding *parC*, most isolates carried a mutation in codon 80 (267 out of 271 isolates). Of these, 112 isolates (42%) harbored a second mutation, located in codon 84. While 40% of the HLR isolates (109 of 271 strains) carried 2 *gyrA* mutations plus one *parC* mutation, the remaining isolates exhibited either additional PMQR genes or mutations in different genomic loci such as *marR* or *acrR*.

Next, the susceptibility profiles of the 553 isolates towards nalidixic acid, ciprofloxacin, norfloxacin and levofloxacin were assessed with the aim to develop a phenotypic algorithm to reliably infer the quinolone resistance mechanisms from disc diffusion results and to predict clinical resistance in *E. coli*. Analyzing the correlation between the results of quantitative disc diffusion testing (i.e., inhibition zone diameters) and genotypes showed that the ECOFF of 19 mm for nalidixic acid achieves the best separation of WT from non-WT isolates (Figure 1 and Appendix A). 

All but 2 strains without any genetically defined resistance mechanisms (138/140, 98.6%) displayed a nalidixic acid-susceptible phenotype, and only 4/411 (1%) isolates with quinolone resistance markers present (2 AAC(6′)-Ib-cr, 1 Qnr-S1 and 1 OqxAB/MarR:Y137H/G103) had inhibition zone diameters above the cut-off value and displayed a susceptible phenotype. We then investigated which fluoroquinolone was the most suitable to discriminate LLR and HLR strains within the nalidixic acid-resistant population. Using a working separator cut-off of 18 mm, the overall agreement between the resistance mechanisms inferred from the observed phenotypes and the WGS-based genotypes for norfloxacin, ciprofloxacin and levofloxacin was 98% (542/553), 98% (542/553) and 96.2% (532/553), respectively (Table 1).

To define possible clinical consequences deriving from the disagreements, i.e., “minor error” (mE), “major error” (ME) or “very major error” (vME), we applied the rule that resistance (phenotypically determined or predicted from WGS data) overrules susceptibility. This rule assumes that if a strain exhibits a susceptible phenotype but carries a resistance mechanism, the error occurs due to the low sensitivity of the phenotypic analysis, and the strain should be thus considered resistant. In this context, even though the resistance gene may be poorly expressed or not expressed at all, activity might be regained in vivo. For these reasons, we deem appropriate classifying these isolates as resistant. Conversely, if an isolate shows a resistant phenotype but no resistance mechanism has been detected, the error is ascribed to the genetic testing, as the disagreement may reflect a yet unknown resistance mechanism; thus, the strain should be considered resistant. Considering these assumptions, phenotype-based algorithms using nalidixic acid in combination with norfloxacin, ciprofloxacin or levofloxacin classified 6, 8 and 11 strains as mE, respectively (Table 1). In sum, nalidixic acid and norfloxacin exhibited the best performance in predicting ECOFF-based WT, LLR and HLR populations, with an excellent agreement between the resistance mechanisms inferred from the observed phenotype and WGS-genotypic data (542/553, 98%). Most importantly, norfloxacin was better than ciprofloxacin in separating LLR and HLR isolates, and discrepancies resulted in only 6 mE (1.1%; Appendix A), which were mostly due to inhibition zone diameters close to the ECOFFs (Appendix A, Appendix A). Altogether, these findings resulted in the establishment of QUIRMIA, given in Figure 2.

### 2.2. EUCAST-Based Classification of Resistance Phenotypes

Next, the correlation between the EUCAST-based phenotypic classification of the isolates and their respective genotypes as determined by WGS was investigated. In general, most diagnostic laboratories infer fluoroquinolone susceptibility by testing ciprofloxacin (as suggested by EUCAST interpretative rules for quinolones and *Enterobacterales* [24]). Therefore, isolates were categorized as susceptible or resistant based on the inhibition zone diameters for ciprofloxacin using the EUCAST-recommended clinical breakpoint of 25 mm. Following one of the recommendations issued by EUCAST in handling ATU values, diameter values falling within the area of technical uncertainty (for ciprofloxacin 22–24 mm) were classified as resistant [25]. Based on this notion, 207 isolates were classified as susceptible and 346 as resistant (Figure 3 and Appendix A). 

Based on WGS data, 138 of the 207 susceptible isolates were categorized as WT (i.e., no resistance mechanism was detected), and 69 isolates were LLR. Of the 346 resistant isolates, 2 were genotypically WT, 73 LLR and 271 HLR. The 142 isolates assigned to the LLR population based on WGS were categorized either as ciprofloxacin susceptible or resistant when analyzed according to EUCAST guidelines (Appendix A). Notably, this population displayed numerous different combinations of resistance determinants. However, despite a different categorization, most isolates (101/142) were genetically uniform and harbored 1 *gyrA* mutation (Table 2).

In sum, classification of *E. coli* isolates using the EUCAST clinical breakpoint of 25 mm for ciprofloxacin reliably separates strains assigned to the wildtype or high-level resistant population by WGS-based analysis, respectively. However, isolates exhibiting a low-level resistance genotype (LLR) may be randomly classified as susceptible or resistant to fluoroquinolones when following EUCAST guidelines.

### 2.3. WGS-Based Prediction of Resistance Phenotypes

Based on the WGS data and the detected resistance mechanism(s) resulting in distinct phenotype(s), the strains were divided into three groups (referred to as resistotypes) (Figure 4). 

Isolates where no resistance mechanism was detected were classified as WT. The concordance between the predicted phenotypes and observed phenotypes was then investigated (Appendix A). We found a high overall agreement between predicted and observed phenotypes (98%) and disagreements resulted in only 5 mE (0.9%). The highest number of discrepant phenotypes was found for strains categorized as LLR (7/142, 4.9%). Again, these discrepancies were mostly due to inhibition zone diameters close to the ECOFFs (Appendix A, Appendix A). Overall, the performance of WGS in predicting ECOFF-based resistance phenotypes was comparable to that of QUIRMIA to infer quinolone resistance mechanisms (Table 1).

Furthermore, the correlation between the WGS-derived classification and the EUCAST-based categorization for ciprofloxacin (Appendix A) was analyzed. Of the 140 isolates where no mechanism was detected, 138 (98.6%) displayed growth inhibition zones equal or above the ciprofloxacin cut-off value of 25 mm and would thus be categorized as susceptible. All 271 isolates classified as HLR based on WGS data had a growth inhibition zone for ciprofloxacin below the cut-off of 25 mm and would thus be categorized as resistant. Notably, the 142 isolates that would be classified as LLR based on WGS-analysis, displayed growth inhibition zone diameters for ciprofloxacin either equal, above or below the cut-off of 25 mm, respectively.

## 3. Discussion

This study provides a comprehensive analysis linking quinolone resistance phenotypes and genotypes using a large dataset of *E. coli* clinical isolates. Based on WGS data, we identified three genetically distinct populations (wildtype, low-level and high-level resistant) that can be reliably discriminated with QUIRMIA using disc diffusion test results for nalidixic acid and norfloxacin and 18 and 19 mm as cut-off separators of 18 and 19 mm, respectively. Notably, nalidixic acid can reliably be used to separate the WT from the non-WT. Further differentiation of the non-WT into LLR and HLR is accomplished using norfloxacin.

Since 2010, EUCAST has discontinued recommending nalidixic acid for the screening of low-level fluoroquinolone resistance. Indeed, EUCAST recommended clinical categorization of isolates based on ciprofloxacin allows for reliable separation of the WT population (i.e., strains without any resistance mechanisms) and isolates displaying high-level resistance (i.e., triple mutants ± PMQR or other genomic mutations). However, ciprofloxacin-based categorization does not allow us to detect LLR, but corresponding isolates are split and either classified as susceptible or resistant. Detection of high-level resistance is critical to prevent treatment failure. In contrast, identification of low-level resistance is crucial in terms of preventing resistance development and may be relevant in situations where bacteria may be exposed to lower, sub-therapeutic antibiotic concentrations, which could eventually promote the emergence of clinical resistance. This may occur in clinical settings due to low-dose prophylactic treatment (such as prior TRUS-guided biopsy or in patients with repeated urinary tract infections), incorrect dosing, poor patient adherence or use of poor quality drugs that do not have the stated amounts of active compound [20,26,27]. Screening for quinolone resistance in fecal samples of patients about to undergo TRUS-guided biopsy, detection of isolates with low-level resistance may help to guide and improve prophylactic treatment. Identification of quinolone low-level resistance could have a clinical impact on patients with recurring urinary tract infections. A recent study has described the detrimental effect of the urinary tract physiological parameters on the activity of ciprofloxacin. This might, in turn, impair the drug’s clinical efficacy against low-level resistant *E. coli* strains and thus favor the emergence of clinically relevant resistance during UTI infections [28]. In sum, the clinical significance of LLR strains remains elusive, and further studies are needed to scrutinize possible in vivo effects. For instance, studies with animal models may contribute to understanding the in vivo effectiveness of quinolones in treating EUCAST-based susceptible and QUIRMIA-based LLR isolates.

PMQR alone or in combination with genomic mutations (such as, e.g., in *soxR* or *acrR*) possibly facilitates the emergence of mutations in the QRDR of *gyrA* and *parC* [7,29] and, eventually, the development of clinical quinolone resistance. In line with previous reports, we show that the use of nalidixic acid for low-level quinolone resistance screening is not efficient in detecting PMQR (such as, e.g., AAC(6′)-Ib-cr, Qnr-S1 and OqxAB) in isolates that otherwise are lacking QDRD mutations [8,11,12]. Further studies are therefore needed to establish susceptibility tests with adequate screening drugs for rapid and reliable recognition of these resistance mechanisms, which can be currently detected only through molecular tests [30].

The advent of next-generation sequencing and a plethora of innovations related to this technology provide a relatively cheap and fast application to collect in-depth genomic data at a large scale. Thus, WGS has been proposed as an alternative to traditional phenotyping for categorization of clinical resistance [31]. However, WGS is neither cost nor time efficient for identification of WT—here, phenotypic tests are still more efficient. In addition, a major drawback of a WGS-based approach remains the knowledge gap regarding genotype/phenotype correlations, which, in turn, impedes the transformation of genomic data into actionable clinical information. In this work, we show that WGS data allow the prediction of ECOFF-based phenotypes and the categorization of isolates as wildtype, low-level and high-level resistant towards fluoroquinolones with an accuracy equal to traditional phenotypic testing. In contrast, when considering the EUCAST-based categorization, we note major genotype/phenotype differences. Then, WGS data allow for a reliable classification of wildtype and high-level resistant isolates as susceptible and resistant to ciprofloxacin, respectively. However, isolates identified as low-level resistant strains in QUIRMIA and WGS are split into different clinical categories, i.e., ciprofloxacin susceptible or resistant, even though they display highly similar or even identical resistance genotypes.

The prevalence of clinical resistance towards fluoroquinolones is notoriously higher in ESBL- and carbapenemase-producing *E. coli* (see also Appendix A) [32,33,34]. In addition, we found that a significant proportion of ESBL and carbapenemase producers (22.7% and 22.2%) would be classified as low-level resistant by the QUIRMIA. Of these, a significant proportion had growth inhibition zones for ciprofloxacin ≥25 mm and would therefore be categorized as susceptible when applying EUCAST criteria. Consequently, failure in detecting low-level resistant quinolone isolates may contribute to the further evolution of multi-drug resistant *E. coli* isolates and even treatment failure.

A limitation of our study is that, although a large has been included, the collection of *E. coli* isolates is biased and reflects the epidemiologic situation of the Zurich region in Switzerland. Further studies with divergent genotypes are warranted to fully evaluate the robustness of QUIRMIA.

In conclusion, we have shown that population-based analysis of phenotypic susceptibility data for nalidixic acid and norfloxacin allows for a reliable and accurate discrimination of three genetically distinct populations, with efficiency comparable to that of WGS. On the other hand, the EUCAST-based classification allows for a coherent categorization of wildtype and high-level resistant isolates, but not of low-level resistant isolates. QUIRMIA provides a coherent genotype/phenotype categorization and may be integrated in the EUCAST expert rule set.

## 4. Materials and Methods

### 4.1. Clinical Isolates

The study included a total of 553 non-duplicate clinical *E. coli* isolates collected in the routine diagnostic laboratory at the Institute of Medical Microbiology (IMM), University of Zurich. The strain collection comprises 440 bacterial strains isolated between February and December 2014 that were used in a previous study [22] and an additional 113 *E. coli* clinical isolates (collected between July 2011 and March 2018) chosen to obtain a broader distribution of susceptibilities towards quinolones. Clinical isolates were considered duplicates and not included in the study if they (i) were obtained from the same patient and (ii) did not exhibit a minimum of three differences in AST interpretation (one major and two minor) with respect to all antibiotics tested as part of the routine diagnostics work-up. The inhibition zone diameters for nalidixic acid, norfloxacin, ciprofloxacin and levofloxacin are listed in Appendix A.

### 4.2. Antimicrobial Susceptibility Testing

Antimicrobial susceptibility testing was performed by the Kirby–Bauer diffusion method according to EUCAST guidelines [35]. Antibiotic discs were from Oxoid Limited, Basingstoke, UK. The inhibition zone diameters were measured with the Sirweb/Sirscan system (i2a) [36]. Routine quality control for AST was performed according to EUCAST guidelines using *E. coli* ATCC 25922 [37].

### 4.3. Whole Genome Sequencing

Bacterial DNA from the isolates was extracted using the DNeasy^®^ Ultraclean^®^ Microbial kit (Qiagen, Hilden, Germany) according to the manufacturer’s instructions. Libraries were prepared with the QUIAGEN QIASeq FX kit (Quiagen, Hilden, Germany) and their quality was analyzed using capillary electrophoresis (Advanced Analytical Technologies Inc., Heidelberg, Germany). DNA libraries were pooled in equimolar concentrations and paired-end sequencing (2 × 150 bp) was carried out on an Illumina MiSeq platform (Illumina^®^, San Diego, CA, USA).

### 4.4. Detection of Resistance Genes

The fastq trimmer tool of the FASTX-Toolkit (Hannon Lab, Cold Spring Harbour Laboratories, Cambridge, UK) was used to filter and trim raw sequencing data with a threshold PHRED score of ≤25. Chromosomal mutations and acquired quinolone resistance genes that are usually located on plasmids were identified using the ARIBA pipeline [38], querying the ARG-ANNOT database and CARD [39,40].

### 4.5. Software

Statistical analysis was performed with the computing environment R (version 3.2.3) [41].

## Figures and Tables

**Figure 1 antibiotics-12-01119-f001:**
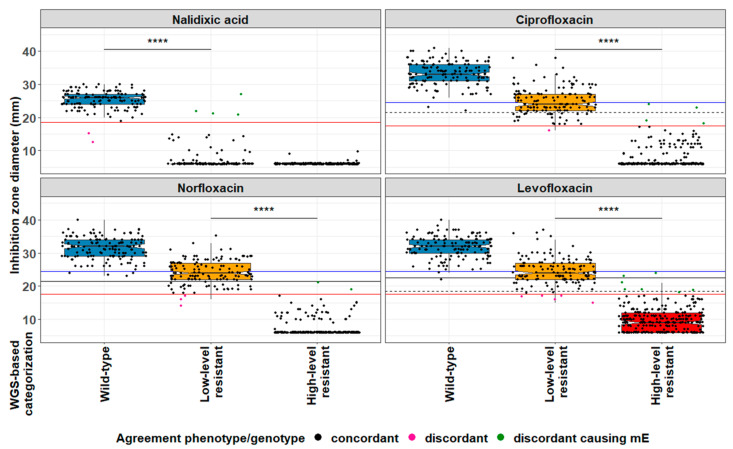
Boxplots of inhibition zone diameters. Lower and upper limits of the boxes represent the first and third quartiles, respectively. The internal horizontal bold lines indicate the median diameters values. Whiskers have a maximum of 1.5 interquartile range. *p* values < 0.0001 are summarized with four asterisks (****). EUCAST lower and upper clinical breakpoints are indicated by the dashed and continuous black lines, respectively, while EUCAST ECOFFs and IMM working cut-off separators are indicated by the blue and red continuous lines, respectively. mE, minor error (i.e., susceptible isolate is categorized as LLR, an LLR isolate as susceptible, an HLR isolate as LLR or an LLR as HLR).

**Figure 2 antibiotics-12-01119-f002:**
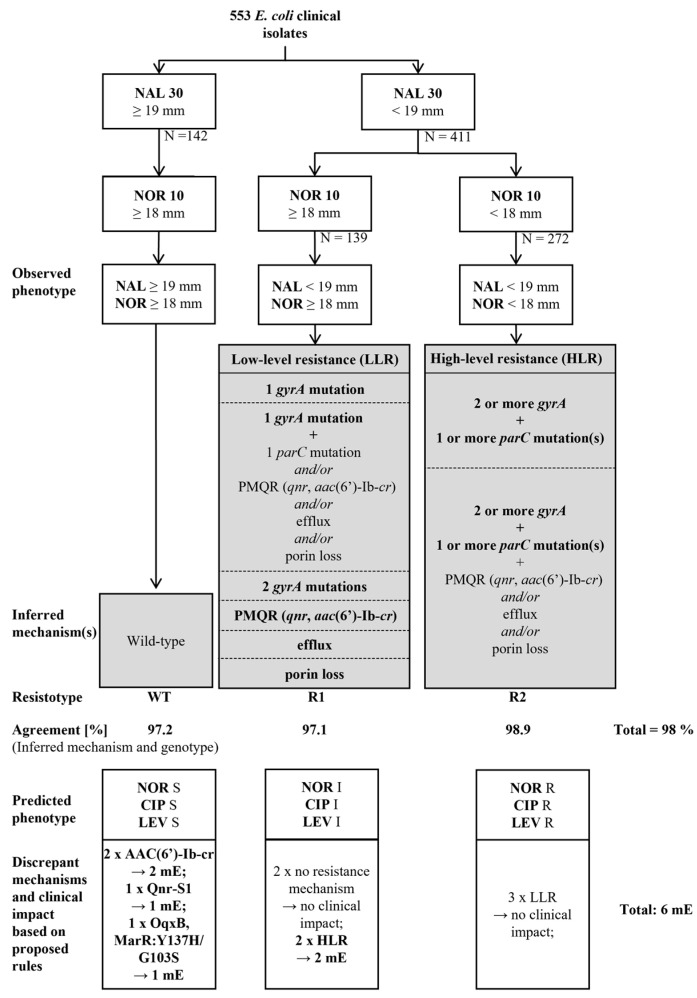
Quinolone Resistance Mechanism Inference Algorithm (QUIRMIA). Clinical isolates were divided based on the inhibition diameters for the quinolones indicated in the nodes. The underlying resistance mechanisms (grouped in resistotypes) were inferred from the observed phenotypes. For each resistotype, the percentage of clinical isolates with concordant QUIRMIA-derived resistance mechanism(s) and WGS-based genotype is indicated. Phenotypes predicted based on the genotype and the clinical impact of the discrepancies are displayed at the bottom of the diagnostic algorithm. NAL, nalidixic acid (30 µg); NOR, norfloxacin (10 µg); R, resistant; I, susceptible by increased dosage; S, susceptible; mE, minor error. Errors are defined as follows: (i) minor error: susceptible isolate is categorized as LLR, an LLR isolate as susceptible, an HLR isolate as LLR or an LLR as HLR; (ii) major error: categorization of a susceptible isolate as HLR (iii) vME: an HLR isolate is categorized as susceptible.

**Figure 3 antibiotics-12-01119-f003:**
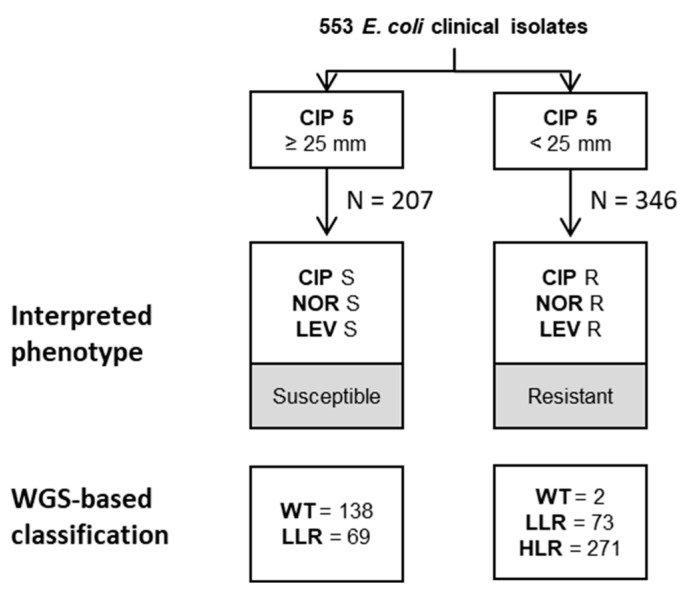
EUCAST-based classification. Strains were categorized as susceptible or resistant to fluoroquinolones based on the inhibition diameters for ciprofloxacin and the clinical breakpoint of 25 mm. In the lowest quadrant, the resistance mechanisms grouped in resistotypes inferred from the NGS data are reported. R, resistant; S, susceptible; CIP, ciprofloxacin; NOR, norfloxacin; LEV, levofloxacin; WT, wildtype; LLR, low-level resistance; HLR, high level resistance.

**Figure 4 antibiotics-12-01119-f004:**
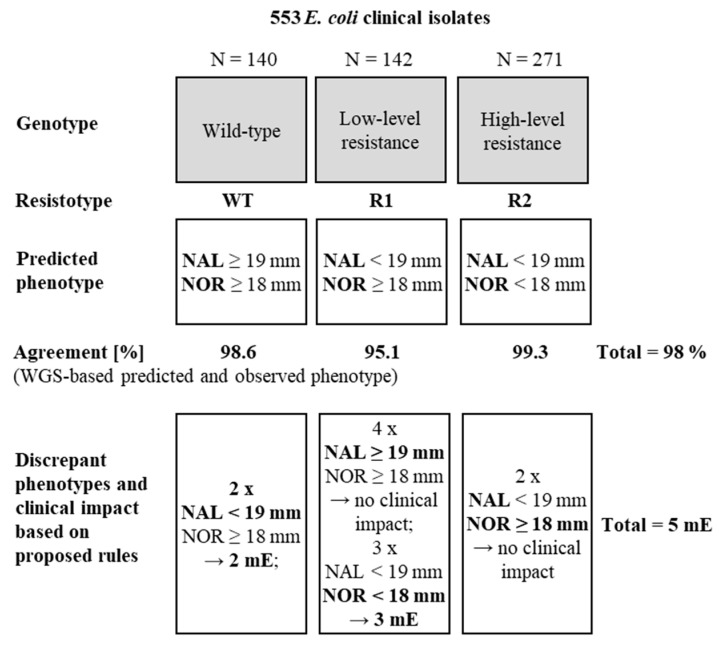
WGS-based quinolone resistance phenotypes. Strains were categorized by resistance mechanism(s) (also referred to as resistotype) based on the WGS data. For each resistotype, the percentage of clinical isolates with concordant predicted and observed phenotype was determined. NAL, nalidixic acid (30 µg); NOR, norfloxacin (10 µg); mE, minor error. Errors are defined as follows: (i) minor error: susceptible isolate is categorized as LLR, an LLR isolate as susceptible, an HLR isolate as LLR or an LLR as HLR; (ii) major error: categorization of a susceptible isolate as HLR (iii) vME: an HLR isolate is categorized as susceptible.

**Table 1 antibiotics-12-01119-t001:** Overall performance of the different algorithms.

Algorithm	Cut-Offs	Number of Isolates Analysed	Wildtype	Low Level Resistance	High Level Resistance	Total	Clinical Impact after Applying Internal Rules ^1^
			Agreement Inferred Mechanisms/Genotype				
**Phenotype-based algorithm**	ECOFF (NAL = 19 mm) ECOFF (NOR = 18 mm)	553	138/142 (97.2%)	135/139 (97.1%)	269/272 (98.9%)	542/553 (98%)	6 mE ^2^
	ECOFF (NAL = 19 mm) ECOFF (CIP = 18 mm)		138/142 (97.2%)	137/143 (95.8%)	267/268 (99.6%)	542/553 (98%)	8 mE
	ECOFF (NAL = 19 mm) ECOFF (LEV = 18 mm)		138/142 (97.2%)	133/142 (91.7%)	264/269 (98.1%)	532/553 (96.2%)	11 mE
**WGS-based algorithm**	ECOFF (NAL = 19 mm) ECOFF (NOR = 18 mm)		**Agreement genotype/phenotype**				
			138/140 (98.6%)	135/142 (95.1%)	269/271 (99.3%)	542/553 (98%)	5 mE ^3^

^1^ Resistance (phenotypically determined or predicted from WGS data) overrules susceptibility. ^2^ See Figure 2. ^3^ See Figure in Section 2.3. CIP, ciprofloxacin; LEV, levofloxacin; NAL, nalidixic acid; NOR, norfloxacin; mE, minor error.

**Table 2 antibiotics-12-01119-t002:** EUCAST-based categorization of the low level resistant (LLR) isolates as determined based on WGS.

Resistance Mechanism(s)	Number of Susceptible Isolates (CIP ≥ 25 mm)	Number of Resistant Isolates (CIP < 25 mm)
1 *gyrA* mutation	52	49
2 *gyrA* mutations	1	3
1 *gyrA* mutation + 1 *parC* mutation	2	7
1 *gyrA* mutation + Qnr-S1	1	4
1 *gyrA* mutation + Qnr-B4	2	-
1 *gyrA* mutation + 2 *marR* mutations	5	-
AAC(6′)-Ib-cr	2	1
Qnr-A1	-	1
Qnr-B4	-	1
Qnr-B19	1	-
Qnr-S1	1	5
Qnr-S2	-	2
Qnr-S1+ 2 *marR* mutations	1	-
OqxB + 2 *marR* mutations	1	-
Total	69	73

WGS, whole-genome sequencing; CIP, ciprofloxacin.

## Data Availability

Whole genome sequencing date were deposited in the European Nucleotide Archive under accession number PRJEB63200.

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
