# Peer review of "QUIRMIA—A Phenotype-Based Algorithm for the Inference of Quinolone Resistance Mechanisms in Escherichia coli"

_antibiotics, 2023, doi:10.3390/antibiotics12071119_

Round 1
Reviewer 1 Report
The manuscript submited by Imkamp et al. presents results of the algorithm developed to evaluate the bacteria resistotype against quinolone antibiotics. The authors studied 553 Escherichia coli isolates based on phenotypic and genetic results of quinolone resistance. The algorithm proved to be easy to understand and applicable to design isolates as wild-type, low-level and high-level quinolone resistance. The attached file contains some suggestions that should be considered.

Author Response
We would like to thank Reviewer 1 for the constructive and valuable comments on the manuscript.
To value the algorithm created (QUIRMIA), I propose that the authors change the title of the manuscript.
The title was modified to emphasize the value of the algorithm created, namely QUIRMIA, as follows:
QUIRMIA – a phenotype-based algorithm for the inference of quinolone resistance mechanisms in Escherichia coli
“This rule assumes that if a strain exhibits a susceptible phenotype but carries a resistance mechanism, the error occurs due to the low sensitivity of the phenotypic analysis, and the strain should be thus considered resistant” I believe that the authors should be careful with this kind of statement, considering that isolates may have the resistance gene but not express it. I suggest revising the sentence.
The paragraph was re-phrased (lines 156-159) as follows: “….the error occurs due to the low sensitivity of the phenotypic test, and the strain should be thus considered resistant. In this context, even though the resistance gene may be poorly expressed or not expressed at all, activity might be regained in vivo. For these reasons we deem appropriate classifying these isolates as resistant.”
In the Materials and Methods section, I believe a topic pertaining to the systematics to create QUIRMIA should be added.
The first part of the Results section comprises a detailed step-by-step description of how QUIRMIA was established, which provides a comprehensive picture regarding the design of QUIRMIA.
What approach was used to determine that these 533 isolates were not duplicates? I believe was determined at the IMM lab. Was this clonality assessed by PFGE or WGS prior to the study? I propose mentioning what was done to make that determination.
Selection criteria are described in the Materials and Methods section (lines 335-339). “Non-duplicates” in this context does not refer to clonality, but i) to the fact that only one isolate per patient was included and that ii) all isolates displayed at least three differences in AST interpretation (two minor and one major) with respect to the drugs tested in our microbiology diagnostic lab (i.e. a panel of at least 16 drugs). The description of the latter criterion in the Materials and Methods section was re-phrased for clarification.
Reviewer 2 Report
The manuscript entitled "Population-based phenotypic inference of molecular resistance mechanisms to provide a coherent genotype/phenotype categorization system for quinolone resistance in Escherichia coli" has an importance in the field of antimicrobial resistance research. The Authors have developed Quinolone Resistance Mechanisms Inference Algorithm a dichotomic decision tree for the inference of quinolone resistance mechanisms from inhibition zone diameters for nalidixic acid and norfloxacin against E. coli. They have showed differences on interpretation of antibiotic resistance status between whole genome sequence and phenotypic method and compared with EUCAST. Furthermore, they add an additional status that is low-level resistance. The manuscript was well written. However, the minor few corrections are required. Please check the uniformity for full form of short form, italic check, word to word space check, figure & table note and others.
Overall background of the study is well written. However, the importance of low-level resistance category needs to be discussed. Materials and method section is short, the Authors may provide more information on the techniques they used.
Line no. 105: Please check the sentence for "low-level (LLR)".
Line no. 332: Please add the full form of AST when using for the first time.
The Authors may add some recommendations or further study plan. Specially, the study should have an animal model to understand the response of EUCAST susceptible and QUIRMIA LLR isolates in animal when they are treated with same antibiotics (based on susceptible status). This data will additionally support the outcome of EUCAST susceptibility and QUIRMIA LLR status.
Author Response
We would like to thank Reviewer 2 for the constructive and valuable comments on the manuscript.
Overall background of the study is well written. However, the importance of low-level resistance category needs to be discussed. Materials and method section is short, the Authors may provide more information on the techniques they used.
In the Discussion section we address to scenarios where the presence of low-level resistant (LLR) isolates may have a detrimental impact on the outcome when quinolones are used for therapy (lines 268-283). Yet, the overall picture regarding the significance of LLR is still blurry and further studies are needed. A corresponding statement has been added to the discussion (lines 283-286), see also your comment regarding the “animal model” below).
We believe that Materials and Methods section provides a concise but comprehensive overview of all relevant technical details.
Line no. 105: Please check the sentence for "low-level (LLR)".
The sentence was edited and split into two (lines 108-109).
Line no. 332: Please add the full form of AST when using for the first time.
Abbreviation was replaced by full form (line 342).
The Authors may add some recommendations or further study plan. Specially, the study should have an animal model to understand the response of EUCAST susceptible and QUIRMIA LLR isolates in animal when they are treated with same antibiotics (based on susceptible status). This data will additionally support the outcome of EUCAST susceptibility and QUIRMIA LLR status.
We agree that further studies are needed to scrutinize and further elucidate the significance of LLR isolates and consider the suggested animal model an interesting idea. A corresponding statement has been added in the Discussion section (lines 283-286).
Reviewer 3 Report
In this study, the authors aim to develop an algorithm, QUIRMIA, for accurately predicting quinolone resistance in E. coli. They found that QUIRMIA effectively inferred resistance mechanisms based on disc diffusion data and genomic sequencing, providing a coherent genotype/phenotype classification. Comparing QUIRMIA with the traditional EUCAST-based classification, QUIRMIA outperformed in identifying low-level resistant isolates. The integration of QUIRMIA into the EUCAST expert rule set could improve outcome prediction and prevent clinical resistance emergence. The study offers valuable insights into quinolone resistance mechanisms and presents a promising algorithm for clinical application. I have only one question: can this algorithm be integrated into a user-applicable tool? Because it may benefit clinical usage.
Author Response
We would like to thank Reviewer 3 for the constructive and valuable comments on the manuscript.
In this study, the authors aim to develop an algorithm, QUIRMIA, for accurately predicting quinolone resistance in E. coli. They found that QUIRMIA effectively inferred resistance mechanisms based on disc diffusion data and genomic sequencing, providing a coherent genotype/phenotype classification. Comparing QUIRMIA with the traditional EUCAST-based classification, QUIRMIA outperformed in identifying low-level resistant isolates. The integration of QUIRMIA into the EUCAST expert rule set could improve outcome prediction and prevent clinical resistance emergence. The study offers valuable insights into quinolone resistance mechanisms and presents a promising algorithm for clinical application. I have only one question: can this algorithm be integrated into a user-applicable tool? Because it may benefit clinical usage.
Yes, we have already integrated this algorithm in our expert system for the interpretation of quinolone disc diffusion results for Escherichia coli. Likewise, this algorithm can be easily implemented in any manual or automated expert system for the categorization of quinolone disc diffusion results for E. coli. Also, we suggested the integration of this algorithm in the EUCAST expert rule set at the end of the manuscript (lines 326-327).
Reviewer 4 Report
The paper "Population-based phenotypic inference of molecular resistance mechanisms to provide a coherent genotype/phenotype categorisation system for quinolone resistant in Escherichia coli" is of much interest and is made to bring new information related the correlation between phenotypic and genotypic interpretation of quinolone susceptibility in E. coli.
In introduction:
lines 74-79: please, explain! If the inhibition diameter zone is between 22-25, means ATU according to EUCAST guideline - which means that the treatment outcome can not be predicted if it is used ciprofloxacin - and they do not recommend to chose either S or R, nor I, but to refrain to interpret the result and even to release the information to the clinician, if there are other antimicrobials that can be used for treatment that specific infection
In results section:
Figure 1: define mE
lines 146-149: "(...) susceptible phenotype but carries a resistance mechanism" - what if that point mutation or the gene that encoding for resistance mechanism does not express? (in this case we'll not be mE but ME). Further studies need to be performed.
The primary objective of an in vitro AST is to predict how a bacterial pathogen may respond to an antimicrobial agent in vivo. The results of the in vitro testing, regardless of method, are interpreted and reported as S, I or R to the action of a particular antimicrobial by applying clinical breakpoints. In Figure 2: at the Observed phenotype - how did you choose NOR limit of 18? Later on on Predicted phenotype is interpreted NOR as S, I or R. According to which guideline? The mentioned one, EUCAST v. 11.0/2021 has other breakpoints and no I interpretation.
Figure 3: weren't no strains in the ATU zone for CIP?
Figure 4: definitions of errors as presented, are according to..? To assert as minor error an HLR isolate as LLR or a LLR as HLR - is not correct according to the accepted errors in testing for antimicrobial susceptibility.
lines 199-202: clinical breakpoint as 25 for CIP does not split strains in S or R, but >=25 equals S and only if <22 is interpreted as R according to the EUCAST version that you mentioned in the paper (same for lines 236-240). And why did you categorised strains with values in ATU as R?
lines 214-216: the LLR as described by you are in the ATU zone according EUCAST guidelines?
Materials and methods:
line 318: when there were isolated those 102 E. coli strains?
lines 321-322: please, explain major and minor differences in AST interpretation
It is mandatory to do quality control when performing AST.
Author Response
We would like to Reviewer 4 for the constructive and valuable comments on the manuscript.
The paper "Population-based phenotypic inference of molecular resistance mechanisms to provide a coherent genotype/phenotype categorisation system for quinolone resistant in Escherichia coli" is of much interest and is made to bring new information related the correlation between phenotypic and genotypic interpretation of quinolone susceptibility in E. coli.
In introduction:
lines 74-79: please, explain! If the inhibition diameter zone is between 22-25, means ATU according to EUCAST guideline - which means that the treatment outcome can not be predicted if it is used ciprofloxacin - and they do not recommend to chose either S or R, nor I, but to refrain to interpret the result and even to release the information to the clinician, if there are other antimicrobials that can be used for treatment that specific infection
Based on EUCAST guidelines, values falling in the ATU can med handled in different ways: the test can be repeated, other methods such as BMD can be performed, the ATU value can be reported without interpretation or as resistant.
As laid out later in the manuscript, we decided to declassify the ATU values as resistant when considering EUCAST based classification of ciprofloxacin disc diffusion values (Figure 3).
In results section:
Figure 1: define mE
The definition of “mE” has been added to the legend of Figure 1 (lines 149-151).
lines 146-149: "(...) susceptible phenotype but carries a resistance mechanism" - what if that point mutation or the gene that encoding for resistance mechanism does not express? (in this case we'll not be mE but ME). Further studies need to be performed.
As also suggested by Reviewer 1, the corresponding paragraph was re-phrased (lines 156-159) as follows: “..the error occurs due to the low sensitivity of the phenotypic test, and the strain should be thus considered resistant. In this context, even though the resistance gene may be poorly expressed or not expressed at all, activity might be regained in vivo. For these reasons we deem appropriate classifying these isolates as resistant.”
The primary objective of an in vitro AST is to predict how a bacterial pathogen may respond to an antimicrobial agent in vivo. The results of the in vitro testing, regardless of method, are interpreted and reported as S, I or R to the action of a particular antimicrobial by applying clinical breakpoints.
In Figure 2: at the Observed phenotype - how did you choose NOR limit of 18?
As laid out in the Result section (lines 139-142), based on the WGS data norfloxacin was the most suitable fluoroquinolone and 18 mm the best cut off to discriminate LLR and HLR strains within the nalidixic acid-resistant population.
Later on Predicted phenotype is interpreted NOR as S, I or R. According to which guideline? The mentioned one, EUCAST v. 11.0/2021 has other breakpoints and no I interpretation.
NOR was interpreted as S, I or R based on the QUIRMIA rules (NAL and NOR cut-off values) and not on clinical breakpoints. Predicted phenotype means how a strain should be classified based on AST data (which may influence therapeutic decision), in this case based on QUIRMIA rules.
Figure 3: weren't no strains in the ATU zone for CIP?
As oulined in the Result section (lines 213-215), when following EUCAST-based classification criteria of disc diffusion data for ciprofloxacin (22-24 mm), ddiameter values falling within the area of technical uncertainty were classified as resistant. In fact, one of the actions recommended by EUCAST in case of ATU values is to report the result as ‘resistant’ https://www.eucast.org/fileadmin/src/media/PDFs/EUCAST_files/Disk_test_documents/ATU/Area_of_Technical_Uncertainty_-_guidance_2019.pdf.
For clarity this information was integrated in the statement.
Figure 4: definitions of errors as presented, are according to..? To assert as minor error an HLR isolate as LLR or a LLR as HLR - is not correct according to the accepted errors in testing for antimicrobial susceptibility.
Based on the notion that LLR (which does not exist in any international guidelines) may be considered as I, we created the definition whereby minor errors occur when a susceptible isolate is categorized as LLR, a LLR isolate as susceptible, a HLR isolate as LLR or a LLR as HLR.
lines 199-202: clinical breakpoint as 25 for CIP does not split strains in S or R, but >=25 equals S and only if <22 is interpreted as R according to the EUCAST version that you mentioned in the paper (same for lines 236-240). And why did you categorise strains with values in ATU as R?
One of the actions recommended by EUCAST in case of ATU values is to report the result as ‘resistant’. https://www.eucast.org/fileadmin/src/media/PDFs/EUCAST_files/Disk_test_documents/ATU/Area_of_Technical_Uncertainty_-_guidance_2019.pdf
On this basis values falling in the ATU for ciprofloxacin were categorized as resistant. This information was added for clarity on lines 213-215.
lines 214-216: the LLR as described by you are in the ATU zone according EUCAST guidelines?
Again, based on the EUCAST classification criteria for disc diffusion values of ciprofloxacin, data falling in the ATU was categorized as resistant. Therefore, LLR isolates exhibiting a low-level resistance genotype (LLR) were classified either as susceptible or resistant to fluoroquinolones.
Materials and methods:
line 318: when there were isolated those 102 E. coli strains?
The additional 113 clinical E. coli strains were isolated between July 2011 and March 2018 at our institution. A corresponding remark has been added (lines 334-335).
lines 321-322: please, explain major and minor differences in AST interpretation
Selection criteria for isolates included in the study were rephrased for clarification (lines 338-340)
It is mandatory to do quality control when performing AST.
We agree. Quality controls for AST are performed on a regular basis in our lab as suggested by EUCAST.
Reviewer 5 Report
The manuscript by Imkamp et al. describes the investigation of molecular mechanisms responsible for quinolone resistance in Escherichia coli and attempts to provide a reliable prediction system for quinolone resistance within this species. The study is scientifically sound and includes more than 500 isolates tested and sequences. The algorithm proposed seems to be very interesting and potentially useful in clinical practice. However, I suggest performing additional validation before it can be published. I have several major and minor comments for the authors.
Major comments:
1. Although the manuscript is dedicated to clinical E. coli isolates, the discussion of quinolone resistance spreading within community population should be made in order to investigate the percentage of true wildtypes. This can be done by analyzing literature data or by scanning the E. coli genomes available in public databases for the mutations providing resistance, which were described in the manuscript.
2. Please describe how you determined the plasmid-mediated quinolone resistance. Since you have not performed long-read sequencing, the approaches to searching for plasmid sequences in WGS data should be described. The presence of some genes usually located on plasmids within genomic data cannot serve as the only evidence in this case.
3. Although the set of the isolates studied was large, it cannot be considered representative since it was taken from previous study and inclusion/exclusion criteria were not set. I suggest performing additional analysis for publicly available genomes, for which the phenotypic data is also available. This will allow performing additional validation of your algorithm and revealing possible incoherence, if any.
4. Please provide the accession or project numbers for the E. coli genomes used in the study. All genomic data should be available in public databases according to the journal rules.
Minor comments:
Citation numbering starts from 4, while 1 and 2 are introduced in the next page – please fix. Citation style does not correspond to journal rules.
p. 2 – please provide a reference (e.g., URL), for Swiss Antibiotic Resistance Report
p. 4 – is “very major errors” a commonly used term? It seems to be ambiguous
p. 4. – QUIRMIA abbreviation has been defined above (p. 3) and does not need to be described again
Author Response
We would like to thank Reviewer 5 for the constructive and valuable comments on the manuscript.
The manuscript by Imkamp et al. describes the investigation of molecular mechanisms responsible for quinolone resistance in Escherichia coli and attempts to provide a reliable prediction system for quinolone resistance within this species. The study is scientifically sound and includes more than 500 isolates tested and sequences. The algorithm proposed seems to be very interesting and potentially useful in clinical practice. However, I suggest performing additional validation before it can be published. I have several major and minor comments for the authors.
Major comments:
- Although the manuscript is dedicated to clinical E. coli isolates, the discussion of quinolone resistance spreading within community population should be made in order to investigate the percentage of true wildtypes. This can be done by analyzing literature data or by scanning the E. coli genomes available in public databases for the mutations providing resistance, which were described in the manuscript.
The major goal of our study was to develop a phenotypic algorithm to provide a coherent genotype/phenotype categorization system for quinolone resistance in E. coli, and not to provide epidemiological information on quinolone resistance in E. coli. Thus, we think that this info would bring no additional value to the study. In addition to that, to address this point we would need epidemiological studies on quinolone resistance in E. coli with large numbers of publicly available genomes of good quality (coverage higher than 20X), which are difficult to find.
- Please describe how you determined the plasmid-mediated quinolone resistance. Since you have not performed long-read sequencing, the approaches to searching for plasmid sequences in WGS data should be described. The presence of some genes usually located on plasmids within genomic data cannot serve as the only evidence in this case.
The term “plasmid-mediated quinolone resistance” refers to resistance genes that – according to literature – are in general found on plasmids. However, in this study we did not perform analyses to determine their location (i.e. chromosomal vs. plasmidic) and we agree that without, it is not possible to claim their origin. This issue was clarified both in the Result (lines 115-117) section as well as in the Materials and Methods section (line 360-361)
- Although the set of the isolates studied was large, it cannot be considered representative since it was taken from previous study and inclusion/exclusion criteria were not set. I suggest performing additional analysis for publicly available genomes, for which the phenotypic data is also available. This will allow performing additional validation of your algorithm and revealing possible incoherence, if any.
We used a large data set of strains covering an extremely broad spectrum of susceptibility profiles for all four quinolones included in the study, as shown from the distributions of inhibition diameters in Figure S1. Thus, we think that adding more isolates would bring no additional value. Moreover, additional validation would require isolates for which both, WGS data of good quality (coverage higher than 20X) and inhibition diameters for nalidixic acid, norfloxacin, ciprofloxacin and norfloxacin are publicly available. This constellation would be very difficult to find, especially because most studies are based on MIC values rather than disc diffusion data.
- Please provide the accession or project numbers for the E. coli genomes used in the study. All genomic data should be available in public databases according to the journal rules.
Genomic data of all isolates used to establish QUIRMIA were deposited at the European Nucleotide Archive (ENA). A corresponding statement and the accession number has been added to the Materials and Methods section (lines 355-356).
Minor comments:
Citation numbering starts from 4, while 1 and 2 are introduced in the next page – please fix. Citation style does not correspond to journal rules.
Citation numbering of the entire document was re-checked and corrected accordingly. Citation style has been adjusted to the journal’s rules.
- 2 – please provide a reference (e.g., URL), for Swiss Antibiotic Resistance Report
The corresponding reference has been added (REF 15)
- 4 – is “very major errors” a commonly used term? It seems to be ambiguous.
‘very major error’ was replaced with the more commonly used abbreviation ‘vME’ throughout the text.
- 4. – QUIRMIA abbreviation has been defined above (p. 3) and does not need to be described again
The sentence has been modified accordingly (line 172).
Round 2
Reviewer 4 Report
Dear authors,
Thank you for the responses to my queries.
I have only one more request: please, add in the Materials and Methods section (4), subsection Antimicrobial susceptibility testing a written information regarding the quality control (type of ATCC strain...). You mentioned that it is obvious that QC for AST was performed, but is needed to be mentioned in the paper, too.
Author Response
We would like to thank Reviewer 4 for further comments.
I have only one more request: please, add in the Materials and Methods section (4), subsection Antimicrobial susceptibility testing a written information regarding the quality control (type of ATCC strain...). You mentioned that it is obvious that QC for AST was performed, but is needed to be mentioned in the paper, too.
As suggested we added the following sentence in the Material and Methods - Antimicrobial susceptibility testing section:
Routine quality control for AST was performed according to EUCAST guidelines using E. coli ATCC 25922 [38].
Reviewer 5 Report
>The major goal of our study was to develop a phenotypic algorithm to provide a coherent >genotype/phenotype categorization system for quinolone resistance in E. coli, and not to provide >epidemiological information on quinolone resistance in E. coli. Thus, we think that this info >would bring no additional value to the study. In addition to that, to address this point we would >need epidemiological studies on quinolone resistance in E. coli with large numbers of publicly >available genomes of good quality (coverage higher than 20X), which are difficult to find.
My comment was not pertinent to performing epidemiological surveillance of E. coli. In your manuscript you claimed the developing of the algorithm for distinguishing wildtype (no resistance) and non-wildtype (acquired resistance). Thus, it is important to estimate the percentage of real wildtypes in population since it could already happened that the spread of wildtypes is low and most isolates have acquired some type of resistance (it can be true, e.g., for K. pneumoniae). There are plenty of studies providing this information for a number of genomes comparable to the one used in your study, e.g., to name a few, 10.1007/s00284-023-03191-6 , https://doi.org/10.1016/j.foodcont.2018.12.043, https://doi.org/10.1186/s12866-023-02796-y. You can just mention the percent of wildtypes revealed by other researchers in the discussion – to my opinion, this will help the readers to understand the applicability of your algorithm.
>We used a large data set of strains covering an extremely broad spectrum of susceptibility profiles >for all four quinolones included in the study, as shown from the distributions of inhibition >diameters in Figure S1. Thus, we think that adding more isolates would bring no additional value. >Moreover, additional validation would require isolates for which both, WGS data of good quality >(coverage higher than 20X) and inhibition diameters for nalidixic acid, norfloxacin, ciprofloxacin >and norfloxacin are publicly available. This constellation would be very difficult to find, >especially because most studies are based on MIC values rather than disc diffusion data.
As far as I understood, you used the dataset described in the manuscript both to derive the classification parameters and to check your results. Even though your dataset is large, it can at the same time be biased since you did not describe the inclusion/exclusion criteria and, of course, could not know the inhibition diameters beforehand. Verification of the algorithm on the independent dataset would demonstrate that it is suitable for real world application and does not reveal the artifacts. However, if suitable reliable independent datasets could not be found currently, I suggest adding the statement describing this to the Discussion.
Author Response
We would like to thank Reviewer 5 for further comments.
My comment was not pertinent to performing epidemiological surveillance of E. coli. In your manuscript you claimed the developing of the algorithm for distinguishing wildtype (no resistance) and non-wildtype (acquired resistance). Thus, it is important to estimate the percentage of real wildtypes in population since it could already happened that the spread of wildtypes is low and most isolates have acquired some type of resistance (it can be true, e.g., for K. pneumoniae). There are plenty of studies providing this information for a number of genomes comparable to the one used in your study, e.g., to name a few, 10.1007/s00284-023-03191-6, https://doi.org/10.1016/j.foodcont.2018.12.043, https://doi.org/10.1186/s12866-023-02796-y. You can just mention the percent of wildtypes revealed by other researchers in the discussion – to my opinion, this will help the readers to understand the applicability of your algorithm.
A good estimate regarding the percentage of real wildtypes requires a relatively large number of isolates reflecting the epidemiologic situation within a given population. Neither the suggested studies nor – to our knowledge - other publicly available studies fulfill this requirement. In addition, we cannot judge the quality of sequencing data. This may have a significant impact on the clinical categorization (wildtype, non-wildtype) e.g., if resistance determinants are not detected due to poor sequence quality. Thus, we prefer not to include any estimates regarding distributions of wildtypes.
As far as I understood, you used the dataset described in the manuscript both to derive the classification parameters and to check your results. Even though your dataset is large, it can at the same time be biased since you did not describe the inclusion/exclusion criteria and, of course, could not know the inhibition diameters beforehand. Verification of the algorithm on the independent dataset would demonstrate that it is suitable for real world application and does not reveal the artifacts. However, if suitable reliable independent datasets could not be found currently, I suggest adding the statement describing this to the Discussion.
Apropriate data sets comprising a relatively large number of isolates, high quality WGS data and disc diffusion data for quinolones were not available when our study was conducted. However, we added as suggested by the reviewer the following statement in the discussion (lines 311 – 314, revised manuscript)
“A limitation of our study is that, although a large dataset has been included in this study, the collection of E. coli isolates is biased and reflects the epidemiologic situation of the Zurich region in Switzerland. Further studies with divergent genotypes are warranted to fully evaluate the robustness of QUIRMIA.”